# Optical Properties and Color Stability of Dental PEEK Related to Artificial Ageing and Staining

**DOI:** 10.3390/polym13234102

**Published:** 2021-11-25

**Authors:** Liliana Porojan, Flavia Roxana Toma, Roxana Diana Vasiliu, Florin-Ionel Topală, Sorin Daniel Porojan, Anamaria Matichescu

**Affiliations:** 1Department of Dental Prostheses Technology (Dental Technology), Center for Advanced Technologies in Dental Prosthodontics, Faculty of Dental Medicine, “Victor Babeș” University of Medicine and Pharmacy Timișoara, Eftimie Murgu Sq. No. 2, 300041 Timișoara, Romania; sliliana@umft.ro (L.P.); flavia.toma@umft.ro (F.R.T.); 2Department of Prosthodontics, Faculty of Dental Medicine, “Victor Babeș” University of Medicine and Pharmacy Timișoara, Eftimie Murgu Sq. No. 2, 300041 Timișoara, Romania; florin.topala@gmail.com; 3Department of Oral Rehabilitation (Dental Technology), Center for Advanced Technologies in Dental Prosthodontics, Faculty of Dental Medicine, “Victor Babeș” University of Medicine and Pharmacy Timișoara, Eftimie Murgu Sq. No. 2, 300041 Timișoara, Romania; porojan.sorin@umft.ro; 4Department of Preventive, Community Dentistry and Oral Health, Faculty of Dental Medicine, “Victor Babes” University of Medicine and Pharmacy Timișoara, 14A Tudor Vladimirescu Ave., 300173 Timisoara, Romania; matichescu.anamaria@umft.ro

**Keywords:** dental polyetheretherketone, optical properties, color stability, artificial ageing, staining

## Abstract

Considering that the processes of PEEK discoloration caused by either intrinsic or extrinsic factors require elucidation, the aim of this study was to investigate the long-term effect of the combined action of ageing and immersing solutions on the optical properties and color stability of PEEK material, related to surface processing (polishing or glazing). (2) Methods: This study aims to determine the influence of different ageing and staining protocols on optical properties, color changes, and surface roughness of a reinforced PEEK material (bioHPP, Bredent, Senden, Germany). For ageing, specimens were submitted to 5000 cycles in a 55 °C bath and a 5 °C bath filled with distilled water. For staining, thermal cycling was performed in a hot coffee bath (55 °C) and a bath filled with distilled water (37 °C) and in a cold juice bath (5 °C) and a bath filled with distilled water (37 °C). Translucency (TP) and opalescence (OP) parameters were determined, the total color change value (ΔΕ*) was calculated, specimens’ surface roughness was analyzed, and statistical analyses were performed. (3) Results: The mean TP values of the studied samples were in the interval of 1.25–3.60, which is lower than those reported for natural teeth or other aesthetic restoration materials. The OP values of PEEK were registered in the range of 0.27–0.75, being also lower than those of natural teeth or other aesthetic restoration materials. OP has a very strong positive relationship with TP. The mean registered Ra values for all subgroups were below 0.13 µm. Artificial ageing and staining in hot coffee proved to increase the roughness values. (4) Conclusions: The glazing of PEEK has a favorable effect on surface roughness and opalescence, irrespective of the artificial ageing or staining protocols. Artificial ageing damages the color stability and roughness of PEEK, regardless of surface processing, and decreases the translucency and opalescence of glazed surfaces. Immersion in hot coffee leads to perceivable discolorations.

## 1. Introduction

The development of thermoplastic high-performance polymers (HPP) has led to an increasing interest in dentistry due to their excellent properties, which has made them suitable for many applications in the field of restorative and prosthetic dentistry. The polyaryletherketone (PAEK) family shows ultra-high mechanical performances and chemical resistance among all the thermoplastic polymers. These materials were introduced to substitute well-known metallic alloys and ceramics for achieving different fixed restorations and removable prostheses [1,2,3].

The term PAEK covers a number of closely related high-performance thermoplastics, like polyetheretherketone (PEEK), polyetherketoneketone (PEKK) and aryl ketone polymer (AKP). Polyetheretherketone (PEEK) is a linear, aromatic, semi-crystalline thermoplastic, developed from bisphenol salts and aromatic dihalides via nucleophilic substitution. The structures of PEEK and PEKK both have aromatic rings, which differ in terms of the ratio of ether and the keto group [4].

PEEK is a material known for its high biocompatibility, favorable mechanical properties, high temperature resistance, chemical stability, high polishing potential, low specific weight, good wear resistance, low plaque affinity, and good bond strength with veneering composites and luting cements. Compared to other materials, such as zirconia, glass ceramics, and metal alloys, PEEK has a low modulus of elasticity (4 GPa), similar to those of bone, and is therefore better from a biomechanical point of view, and it absorbs destructive fracture energy and functional stresses, reducing the forces transferred to the abutment teeth. PEEK composites reinforced with carbon fibers (CFR-PEEK) have a higher elastic modulus (18 GPa). PEKK shows better mechanical properties in terms of flexure, tensile, and compressive strength [2,5,6,7,8,9,10,11,12].

Reinforcing agents may be used in order to absorb and transfer occlusal forces from the weak polymeric matrix to the stronger filler material [1,13,14]. HPP are expected to withstand occlusal loads during functions and consequently have to display mechanical strength, in order to prevent cracks, fractures, plastic deformation, or even failures [12,15].

The thermal properties of dental polymers are essential for predicting their lifetime performances, developing adequate processing protocols, and monitoring the effects of ageing [16]. After ageing in different solutions, PEEK had lower solubility and water absorption values compared to composite resins, hybrid materials, and PMMA-based materials [17].

Veneering or glazing is challenging due to the stable chemical structure and the unreactive and inert surface characteristics of HPP [18,19,20,21]. 

Even though PEEK materials are the aesthetic alternative of metallic frameworks, due to the grayish-brown or pearl-white opaque color, frameworks of fixed restorations have to be veneered with a composite resin. If the veneer is only partial, a part of the framework remains visible in the oral cavity. In addition, when using the material for the removable partial denture frameworks, their compounds, either connectors, or direct retainers, clasps, attachments, or double crowns, will be exposed in the oral cavity. Therefore, the surface processing and finishing should be a challenge. Studies are needed in order to determine optical properties and color stability during artificial ageing and staining. 

The discoloration of dental prostheses may be caused by either intrinsic factors, such as the type of resin matrix, percentage and filler size, composition and polymerization mode, chemical reactions within the restorative material, age, and restoration processing mode, or extrinsic factors, such as staining from food colorants, such as caffeine, theobromine, anthocyanidins, tannins, and nicotine, in drinks, beverages, mouth rinses, and smoking [22,23,24,25].

The processes of PEEK discoloration still remain to be elucidated. Color stability significantly depends on surface roughness and surface-free energy during surface processing. Several studies showed that there is a positive correlation between a high surface roughness and the discoloration of denture resins, explained by the larger contact area [26,27,28].

The aim of this study was to investigate the long-term effects of the combined actions of ageing and immersing solutions on the optical properties and the color stability of PEEK material, related to surface processing (polishing or glazing). The null hypotheses were: (a) surface processing by glazing has a positive effect on the optical properties, color stability, and surface roughness, (b) artificial ageing has no effect on the optical properties, color stability, and roughness, and (c) immersion in staining beverages leads to discolorations.

## 2. Materials and Methods

This study examined the influence of different ageing and staining protocols on optical properties, color changes, and surface roughness of a reinforced PEEK material (bioHPP, Bredent, Senden, Germany) The material was composed in 20% ceramic fillers with a size of 0,3 -0,5 microns [29] (Table 1). 

### 2.1. Specimen Preparation

PEEK blanks were sliced in rectangular-shaped plates with a thickness of 1 mm. They were polished using silicon carbide papers (600–2000 grit) and the final thickness of each specimen was checked with a digital caliper. The specimens were finally manually polished with a low-speed handpiece and diamond polishing paste, Renfert Polish (Renfert, Hilzingen, Germany), ultrasonically cleaned for 10 min, degreased with 98% ethylic alcohol, and dried. Specimen surfaces were then divided into 2 groups (n = 40) in terms of the applied surface treatment method: conventional polishing (p) and glazing after polishing (g). Resin Glaze Primer (Shofu, Kyoto, Japan) was applied to the PEEK surfaces for 60 s and allowed to dry, and then two thin layers of glaze, Resin Glaze Liquid (Shofu, Kyoto, Japan), were applied with a soft brush, in one direction to eliminate air bubbles, and were polymerized for 180 s each in a light-polymerizing device, SIBARI Polymerizer SR620 (Sirio Dental, Meldola, Italy), with a wavelength between 400–560 nm. The specimens of each group were randomly divided into five subgroups (A, B, C, D, E) (n = 8) consisting of different protocols. The specimens of subgroup A represent the control group (Table 2). 

### 2.2. Ageing Protocol

Specimens of subgroups B–E were stored in distilled water at 37 °C for 7 days. Specimens of subgroups C–E were subjected to 5000 cycles in a 55 °C bath and a 5 °C bath filled with distilled water. Each cycle lasted 80 s: 30 s in a 55 °C bath, 10 s to transfer the specimens to the other bath, 30 s in a 5 °C bath, and 10 s to transfer the specimens back to the 55 °C bath.

### 2.3. Staining Protocol

Specimens of subgroups D and E were subject to different staining protocols: thermal cycling in a hot coffee bath (55 °C) and a bath filled with distilled water (37 °C) (D) and thermal cycling in a cold juice bath (5 °C) and a bath filled with distilled water (37 °C), respectively (E). Each cycle lasted 80 s: 30 s in a 55 °C or 5 °C bath, 10 s to transfer the specimens to the other bath, 30 s in a 37 °C bath, and 10 s to transfer the specimens back to the 55 °C or 5 °C bath. A total of 720 cycles were used for simulating 2 min per day of contact with the respective beverages. The instant coffee solution consisted of 1.8 g of instant coffee powder (Nescafe Brasero, Nestle, Vevey, Switzerland) per 150 mL of boiling water. Coca-Cola juice was used as cold, carbonated soft beverage (Coca-Cola Company, Atlanta, GA, USA).

### 2.4. Optical Measurements and Color Change Determinations

Translucency and opalescence parameters were determined for all specimens before and after thermal cycling. Optical properties were calculated under a D65 illuminant, using a spectrophotometer, Vita Easyshade IV (Vita Zahnfabrick, Bad Säckingen, Germany). The spectrophotometer was calibrated before each measurement.

Black (b) and white (w) backgrounds were used to assess the measurements, using a grey card, WhiBal G7 (White Balance Pocket Card). L* is a measure of the lightness–darkness of material (perfect black has an L* = 0, and perfect white has an L* = 100). The a* coordinate represents the redness (positive value) or the greenness (negative value), while the b* coordinate is a measure of the yellowness (positive value) or the blueness (negative value) [30,31,32].

TP values were calculated using Equation (1):TP = [(L_b_ − L_w_)^2^ + (a_b_ − a_w_)^2^ + (b_b_ − b_w_)^2^]^1/2^(1)

OP values were calculated using Equation (2):OP = [(a_b_ − a_w_)^2^ + (b_b_ − b_w_)^2^]^1/2^(2)

CR was achieved by Equation (3):CR = Y_b_/Y_w_ Y = [(L* + 16)/116]^3^ × 100 (3)
w and b are color coordinates of the specimens on the white and black backgrounds. In this calculation, CR = 0 is considered transparent, and CR = 1 is regarded as totally opaque [33].

The color changes (ΔE*) were calculated based on the CIE L*a*b*color system. L* represents lightness (+ bright, and − dark), a* represents the color scale from red (+) to green (−), and b* represents the color scale from yellow (+) to blue (−).

The total color change value (ΔΕ*) was calculated according to Equation (4), which represents the color difference before and after immersion:ΔΕ* = [(ΔL*)^2^ + (Δa*)^2^ + (Δb*)^2^]^1/2^(4)

Measurements were made for each group.

The National Bureau of Standards (NBS) system was used to quantify the levels of color change (Table 3). To relate the color change to a clinical standard, the ΔE* values were converted into NBS units: NBS = ΔE* × 0.92 [34,35,36,37,38].

### 2.5. Surface Roughness Measurements

Specimens’ surface roughness was analyzed with a contact 2 µm stylus profilometer, Surftest SJ-201 (Mitutoyo, Kawasaki, Japan), for each subgroup. Arithmetic average roughness (Ra) and maximum absolute vertical roughness (Rz) measurements were performed in 5 different directions and all data were recorded. The mean value of the 5 measurements was calculated for each surface. The sampling length was 0.8 mm, and a force of 0.7 mN was applied. 

### 2.6. Statistical Analysis

Statistical analyses were performed by means of the IBM SPSS Statistics software (IBM, New York, NY, USA). The differences among the variables were noted. Average values and standard deviations (SD) were calculated. A paired t test was used to evaluate the comparisons between the means. A p value of under 0.05 was considered statistically significant. Spearman correlation was used to assess monotonic similar or dissimilar relationships (whether linear or not) between variables. It measures the strength of association between variables and the direction of the relationship. The significance was related to: 0.00–0.19—“very weak”, 0.20–0.39—“weak”, 0.40–0.59—“moderate”, 0.60–0.79—“strong”, and 0.80–1.0—“very strong”.

## 3. Results

L*, a*, b* values were registered on white and black backgrounds, and the calculated TP, CR and OP values are displayed in Figure 1, Figure 2 and Figure 3 for each subgroup.

The mean values within groups exhibited significant differences between Ag and Cg (for TP and OP parameter) and between Ap and Bp (for CR parameter ) (Table 4). This means that the significant differences of optical properties between hydrated polished samples and the control group, and thermal cycled glazed samples and the control group, are registered. 

Related to the differences between the polished and glazed samples, *p* values are unsignificant for TP (*p* = 0.120) and CR (*p* = 0.069), but significant for OP (*p* = 0.034), meaning that glazed subgroups are more opalescent than polished ones, both in the control subgroups, and after ageing and staining. 

The correlations are positive and strong between TP and CR (r = 0.733) and CR and OP (r = 0.624) and very strong between TP and OP (r = 0.939).

According to NBS units, perceivable color changes were calculated for ageing thermal cycled groups and groups thermal cycled in coffee (Figure 4). Related to surface processing, between polished and glazed samples, the differences were not significant (*p* = 0.061).

Mean roughness values Ra and Rz with SD values are included in Figure 5 and Figure 6.

Statistical analyses show significant Ra differences between the control group and the group aged by thermocycling and stained by thermocycling in coffee. The glazed control group shows additional significant differences with the group stored in water. Ageing and thermal cycling in coffee increased the Ra values (Table 5). Between Ra and Rz is a very strong positive correlation (r = 0.915). Between the polished and glazed groups, the roughness differences are significant (*p* = 0.020).

There is a very strong positive correlation (0.976) between the surface roughness (Ra) and the color stability (∆E) for the glazed samples and positive strong correlation (0.634) for the polished samples. 

The first hypothesis, (a) “surface processing by glazing has a positive effect on the optical properties, color stability, and surface roughness”, is partially accepted; glazing has a positive significant effect on OP values from the optical properties, and on the surface roughness, but not on color stability.

The second hypothesis, (b) “artificial ageing has no effect on the optical properties, color stability, and roughness”, is rejected; it has a negative effect on the TP and OP values of glazed samples, and a negative effect on color stability and roughness for all samples.

Hypothesis (c), “immersion in staining beverages leads to discolorations”, is partially accepted; only samples thermal cycled in hot coffee registered perceivable color changes.

## 4. Discussion

Computer-aided design (CAD) and computer-aided manufacturing (CAM) technology enables processing PEEK by milling from prepressed blanks and is the most used at present. Related to the processing technology, studies proved that milled fixed and removable restorations showed appropriate mechanical properties. The use of different surface processing and finishing methods may be a subject of interest, in terms of the effect of the oral environment on the optical properties and color stability [3,12,39,40].

The optical properties of dental materials, such as translucency and opalescence, are critical factors for aesthetics and to mimic the natural appearance of the restorations. Translucency (TP) is defined as the difference in color at a uniform thickness measured by a spectrophotometer device over white and black backings. If the material is completely opaque, the TP value is zero. As the TP value increases, the translucency of the material also increases. Opalescence is defined as a scattering of wavelengths of visible light, and as a result, an object appears bluish and orange/brown in the reflected and transmitted color [29,41,42,43,44,45,46].

Translucency can be expressed as the relative amount of light passing through the unit thickness of a material. With respect to the factors which influence the translucency, the surface conditions, water content, and illumination were reported. Surface finishing procedures could modify surface topography and consecutive light scattering. As a result, surface finishing significantly affects translucency. The mean TP values of 1 mm-thick bovine enamel and dentine and human enamel and dentine were 14.7, 15.2, 18.7, and 16.4, respectively. The adjustment of the translucency of aesthetic dental restorative materials has been investigated, by the influence of filler size, and the difference between the transmitted, reflected colors and the translucency of experimental restorative materials were determined [47,48,49]. For zirconia, TP values were registered between 9.10 and 4.83 [50].

In addition to the effect of filler in dental composite materials, the effects of thermocycling and staining were found to influence TP values [51,52,53,54,55,56]. Even if PEEK materials belong to types not used for monolithic restorations, they are an aesthetic alternative to metallic frameworks. Therefore, their optical properties are important and have to be related to the optical features of teeth. The mean TP values of the studied samples were in the interval of 1.25–3.60, which is lower than those reported for natural teeth or other aesthetic restoration materials.

The CR and TP values of aesthetic dental materials were compared in different studies. As a result, CR varies inversely to TP (correlation coefficient: r = −0.93). The mean CR values of human and bovine enamel and dentine were negatively correlated with TP values (r = −0.93 to −0.78) [47,57,58]. This study shows a positive strong correlation between TP and CR, which is proof that the optical properties are not similar to those of natural teeth structures.

The OP value of the enamel–dentin complex was reported to be 4.8, and that of enamel, 7.4 [59,60]. The opalescence of tetragonal zirconia has been reported to be in the range of 1.25 to 2.83 [61]. The OP values of PEEK were registered to be in the range of 0.27–0.75, which is also lower than those of natural teeth or other aesthetic restoration materials.

OP is strongly positively correlated to CR and very strongly positively correlated to TP. The glazed samples proved to be more opalescent compared to the polished ones. A possible reason for this could be due to the glaze layers that increase the yellowness of the samples [62]. The behavior of the aesthetic dental restorations during their clinical use is essential and in vitro studies may simulate the oral environment, based on the type of material and processing method, in order to achieve reliable restorations. For the estimation of the long-term clinical stability of dental restorative materials, studies conducting thermal cycling tests to simulate oral environmental variables have been conducted [29,63,64]. Thermal cycling is a useful method to accelerate the artificial ageing of the samples. This is useful because it can estimate the clinical performance by reproducing the temperature in the oral environment, which contributes to long-term degradation [65]. The water ageing method includes standardized thermal variations with baths ranging from 5 to 55 °C for several cycles [66]. The thermal ageing protocol was involved in the artificial ageing simulation with 5000 cycles, simulating a clinical period of 6 months [65]. On the other hand, the staining cycling period has been equated with an immersion of 2 min/day, meaning 360 min/6 months, respective to 720 cycles for each sample and each beverage.

As the color of the restoration is also affected by the surface roughness, the roughness parameters were also taken into consideration in this study. A rough surface reflects less light than a smooth surface, altering the optical properties. In addition to the optical properties, increased surface roughness values adversely affect the strength of the materials and, implicitly, that of the restorations. Different surface roughness parameters are used to measure surface roughness, recording the highest peaks and lowest valleys of the surface profile [67,68,69,70]. Average roughness (Ra) is the most frequently used. Ra values below 0.2 μm are generally clinically accepted [71,72,73]. The mean registered Ra values for all subgroups were below 0.13 µm. Artificial ageing and thermal cycling in hot coffee proved to increase the roughness values. In this study, the maximum absolute vertical roughness (Rz) was also registered and a strong positive correlation was found. A possibility for the increased surface roughness after hot coffee immersion can be due to the absorption of water and the solubility of the colorants. Coffee contains a high number of molecular-weight, water-soluble colorants [74].

## 5. Conclusions

Within the limitations of this laboratory study, the following conclusions can be drawn:The glazing of PEEK has a favorable effect on surface roughness and opalescence, irrespective of the artificial ageing or staining protocols.Artificial ageing damages the color stability and roughness of PEEK, regardless of surface processing, and decreases the translucency and opalescence of glazed surfaces.Immersion in hot coffee leads to perceivable discolorations.

## Figures and Tables

**Figure 1 polymers-13-04102-f001:**
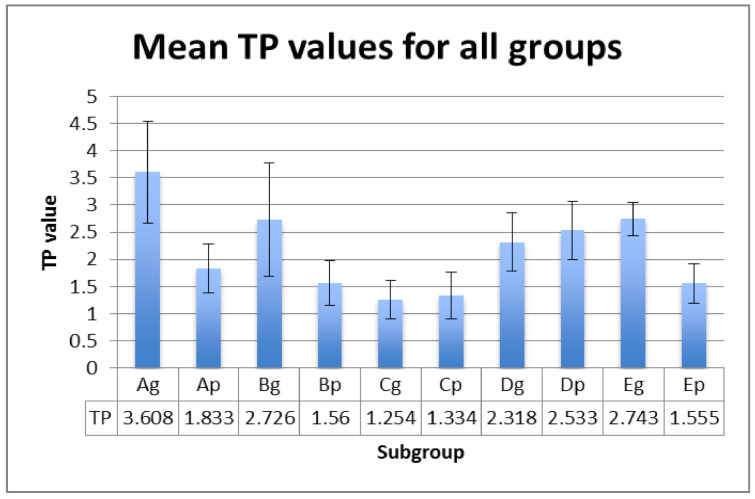
TP values (mean ± SD) for all tested groups.

**Figure 2 polymers-13-04102-f002:**
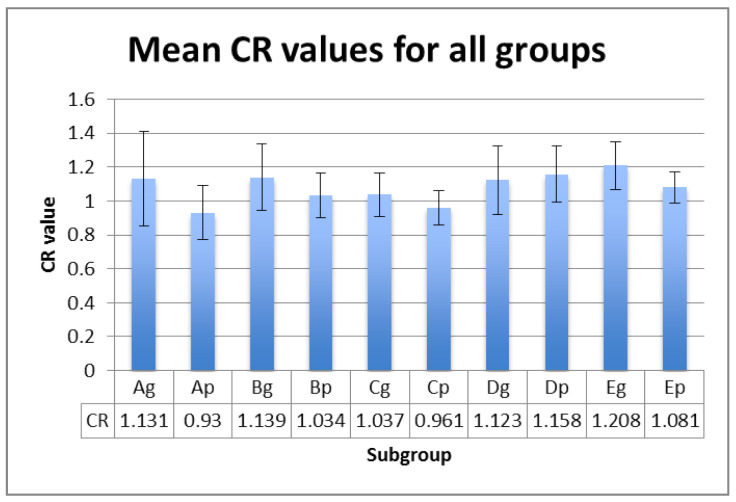
CR values (mean ± SD) for all tested groups.

**Figure 3 polymers-13-04102-f003:**
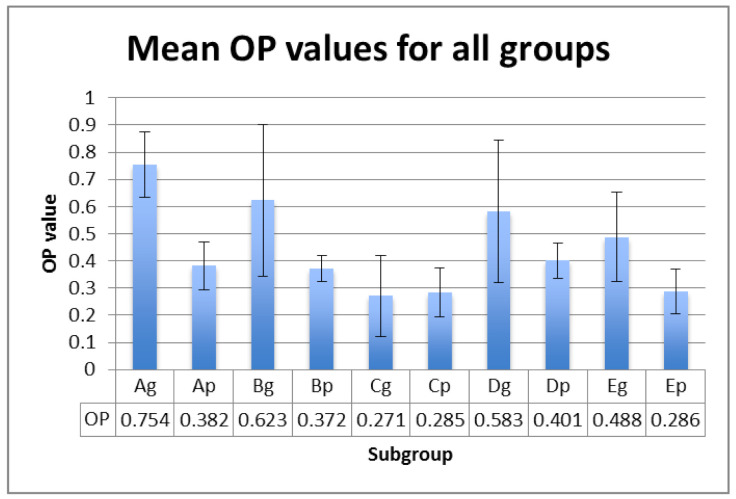
OP values (mean ± SD) for all tested groups.

**Figure 4 polymers-13-04102-f004:**
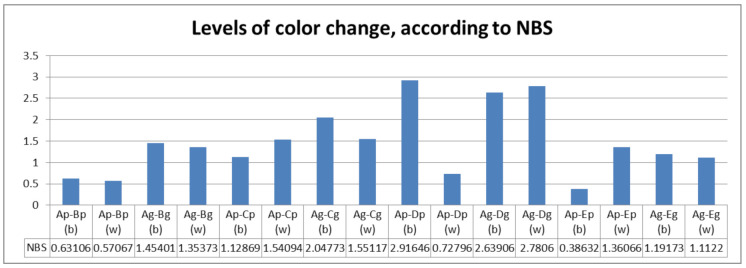
Levels of color change for each subgroup, according to NBS. b(black), w(white).

**Figure 5 polymers-13-04102-f005:**
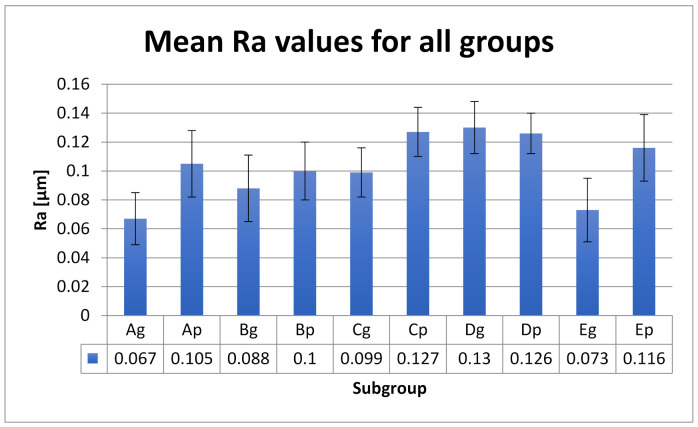
Ra values (mean ± SD) for all tested groups.

**Figure 6 polymers-13-04102-f006:**
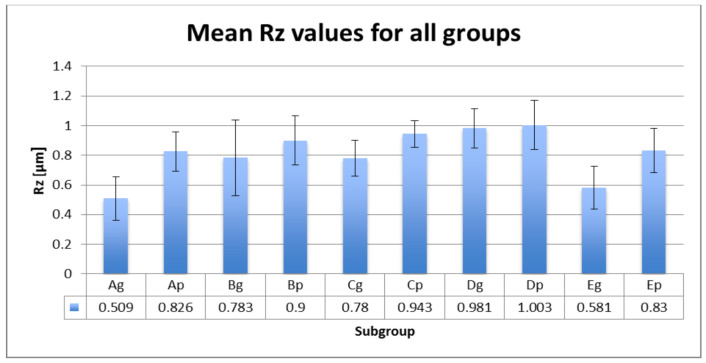
Rz values (mean ± SD) for all tested groups.

**Table 1 polymers-13-04102-t001:** The composition of the studied material.

Name of the Material	Composition
BioHPP, Bredent, Senden, Germany	20% ceramic fillers in a high-performance polymer

**Table 2 polymers-13-04102-t002:** Protocols according to each subgroup.

Subgroup	Processing Protocol
Ap, Ag	Control
Bp, Bg	Storing in distilled water for 7 days
Cp, Cg	Storing in distilled water + thermal cycling
Dp, Dg	Storing in distilled water + thermal cycling + hot coffee staining
Ep, Eg	Storing in distilled water + thermal cycling +cold juice bath (5 °C)

**Table 3 polymers-13-04102-t003:** Levels of color change, according to NBS.

NBS Units	Color Changes
0.0–0.5	extremely slight change
0.5–1.5	slight change
1.5–3.0	perceivable
3.0–6.0	marked change
6.0–12.0	extremely marked change
12.0 or more	change to another color

**Table 4 polymers-13-04102-t004:** *p* values between groups, related to optical properties.

Comparison of Optical Properties Values between Groups	*p* Value for TP	*p* Value for CR	*p* Value for OP
Ap–Bp	0.534	0.045	0.891
Ap–Cp	0.328	0.594	0.352
Ap–Dp	0.199	0.018	0.864
Ap–Ep	0.495	0.050	0.153
Ag–Bg	0.323	0.953	0.453
Ag–Cg	0.004	0.433	0.000
Ag–Dg	0.086	0.944	0.212
Ag–Eg	0.427	0.558	0.067

**Table 5 polymers-13-04102-t005:** *p* values between groups, related to Ra.

Comparison of Ra Values between Groups	*p*-Value
Ap–Bp	0.409
Ap–Cp	0.005
Ap–Dp	0.014
Ap–Ep	0.205
Ag–Bg	0.005
Ag–Cg	0.000
Ag–Dg	0.000
Ag–Eg	0.454

## Data Availability

Not applicable.

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
