# Peer review of "Optical Properties and Color Stability of Dental PEEK Related to Artificial Ageing and Staining"

_polymers, 2021, doi:10.3390/polym13234102_

Round 1
Reviewer 1 Report
This study investigated color stability and surface roughness of the polished and grazed PEEK treated with ageing, thermocycling, and staining process. The results clearly indicated and of interest in dental materials field. I recommended this paper to accept the Journal after a revision. I commented on the content of present paper as follows.
- Examined PEEK
In Materials and Methods section, detail information for the PEEK should be presented, e. g. brand name, company, country, filler content, etc. This information helps to understand the results for the PEEK. I recommend addition of new table for the information.
- Typo in Table 3
In the column for p value for TA, “TA” may be typo. TP is correct?
- Correlation
Is there any statistical correlation between color stability and surface roughness? Please clarify this.
- Discussion
In line 382, “Artificial ageing and thermal cycling in hot coffee proved to increase the roughness values.” Please discuss the reason based on material science. What happen on the material surface?
- Discussion
As similar please discuss about the reason for the sentence in line 248–251 “Related to differences between ~, and after ageing and staining.”
Author Response
Thank you for the review!
This study investigated color stability and surface roughness of the polished and grazed PEEK treated with ageing, thermocycling, and staining process. The results clearly indicated and of interest in the dental materials field. I recommended this paper to accept the Journal after a revision. I commented on the content of present paper as follows.
- Examined PEEK
In Materials and Methods section, detail information for the PEEK should be presented, e. g. brand name, company, country, filler content, etc. This information helps to understand the results for the PEEK. I recommend addition of new table for the information.
Response: We agree to this and a new table was added.
The material is composed in 20% ceramic fillers with a size of 0,3 -0,5 microns [73] (Table 1).
Table 1. The composition of the studied material.
|
Name of the material |
Compositon |
|
BioHPP, Bredent, Senden, Germany |
20% ceramic fillers in a high performance polymer |
2.Typo in Table 3
In the column for p value for TA, “TA” may be typo. TP is correct?
Response: We agree to this point and the changes were made. We corrected- TP.
- Correlation
Is there any statistical correlation between color stability and surface roughness? Please clarify this.
Response : There is a correlation between color stability and surface roughness. Ra-∆E 0.976 (very strong positive )glazed samples and for the polished 0,634 (strongly positive ).
- Discussion
In line 382, “Artificial ageing and thermal cycling in hot coffee proved to increase the roughness values.” Please discuss the reason based on material science. What happen on the material surface?
Response: A possibility for the increased surface roughness after hot coffee immersion can be due to the absorbtion of water and the solubility of the of the colorants. Coffee contains a high number of molecular weight water solubale colorants [74].
- Discussion
As similar please discuss the reason for the sentence in lines 248–251 “Related to differences between ~, and after aging and staining.”
Response: We agree to this point. Additional information was added to the manuscript.
The glazed samples proved to be more opalescent compared to the polished ones. A possibility could be due to the glaze layers that increase the yellowness of the samples [75].
Reviewer 2 Report
Several minor revision suggestions:
- In title, “Peek” should be “PEEK”
- Line 120, it is better to provide the light intensity of the curing device
- Some data, e.g. Figs. 1&3 have very large coefficients of variations as the length values of error bars are very high. Is there any way to improve? At least the authors should provide some explanation in the manuscript.
- Regarding the term “p-value”, upper-case P and lower-case p are both used in different publications. However, the authors should only keep one choice throughout one manuscript. In this manuscript, the authors use lower-case “p” in most cases while upper-case “P” also appears such as in Table 4.
Author Response
Thank you for the review!
In title, “Peek” should be “PEEK”
Response: We agree to his point and the changes were made. The title is now : Optical Properties and Color Stability of Dental PEEK Related to Artificial Ageing and Staining.
Line 120, it is better to provide the light intensity of the curing device
Response : We agree to this and additional informations were added in article. The light intensity for the light curing device was in 400-560 nm.
Some data, e.g. Figs. 1&3 have very large coefficients of variations as the length values of error bars are very high. Is there any way to improve? At least the authors should provide some explanation in the manuscript.
Response: We agree to this point and the figures 1-3 were changed accordingly. We changed the SD values in the figures as well as requested.
Regarding the term “p-value”, upper-case P and lower-case p are both used in different publications. However, the authors should only keep one choice throughout one manuscript. In this manuscript, the authors use lower-case “p” in most cases while upper-case “P” also appears such as in Table 4.
Response :We agree to this point and we changed the manuscript with a lower case p-value.